# Relationship between ultra-widefield optical coherence tomography and ophthalmoscopy for detecting posterior inflammation in posterior uveitis and panuveitis

**Norihiko Misawa, Mizuki Tagami●\*, Atsushi Sakai, Yusuke Haruna, Shigeru Honda**

Department of Ophthalmology and Visual Sciences, Graduate School of Medicine, Osaka Metropolitan University, Osaka, Japan

\* mizuki1979feb@yahoo.co.jp

## Abstract

### Purpose

To confirm the utility of ultra-widefield optical coherence tomography (W-OCT) for diagnosing uveitis.

### Method

We retrospectively studied patients who had been diagnosed with uveitis and had undergone W-OCT. All patients had visited at Osaka Metropolitan University between January 2019 and January 2022. On W-OCT, vitreous opacity ("W-OCT VO") and the presence of vitreous cells ("W-OCT Cells") were identified by three specialists. We compared findings from ophthalmoscopy ("Ophthalmoscopic findings") and fluorescein angiography ("FAG findings") with those from W-OCT.

### Results

This study investigated 132 eyes from 68 patients (34 males, 34 females; mean age, 53.97 ±22.71 years). Vitreous cells in posterior uveitis and panuveitis differed significantly between "W-OCT Cells" and "Ophthalmoscopic findings" for all cases ($P = 0.00014$). Vitreous opacities in posterior uveitis and panuveitis did not differ significantly between "W-OCT VO" and "Ophthalmoscopic findings" ($P = 0.144$) for all cases. Compared to "Ophthalmoscopic findings", "W-OCT Cells" offered 51.1% sensitivity and 66.7% specificity for all cases ($p<0.01$). Compared to "Ophthalmoscopic findings", "W-OCT VO" offered 78.6% sensitivity and 30% specificity for all cases ($p = 0.19$). In addition, "W-OCT Cells" did not differ significantly from "FAG findings" for all cases ($P = 0.424$).

### Conclusion

W-OCT was shown to offer significantly greater sensitivity than ophthalmoscopy for detecting vitreous cells. The results of this study may add an option for the evaluation of uveitis.

**Data Availability Statement:** All relevant data are within the paper and its Supporting Information files.

**Funding:** The authors received no specific funding for this work.

**Competing interests:** The authors have declared that no competing interests exist.

## Introduction

Uveitis is an inflammatory disease that threatens sight worldwide [1]. The annual number of new cases of uveitis is reportedly 17–52 per 100,000 persons [2–4]. The quantification of inflammatory findings has been used to diagnose uveitis since 1959 [5]. The diagnosis of uveitis also requires a physical examination and serological tests [6]. A variety of tests are available, however standardizing testing methods will lower the cost of testing [7]. Tests that can be standardized are therefore important. In recent years, "The SUN* Working Group Grading Scheme for Anterior Chamber Cells" scale [8] defined by the SUN Working Group are often used. This scale allows the quantification of ophthalmoscopic findings.

In ophthalmology, ophthalmoscopy is a fundamental examination. The technology of ophthalmic diagnosis has evolved due to the appearance of optical coherence tomography (OCT). In uveitis, OCT allows the identification of changes in retinal shape such as macular edema [9]. OCT is an imaging modality that also allows objective evaluation through analysis. However, from the perspective of technical feasibility, issues such as examination time mean that OCT can only evaluate a portion of the retina, such as the macula. Technological developments have increased the scanning speed of OCT and have thus allowed scanning of a wider angle. In the past, OCT has been reported to allow direct visualization of vitreous inflammatory cells in patients with uveitis [10]. However, such evaluations have only been performed for the macular area. The Xephilio OCT-S1 (Canon, Tokyo, Japan) is an ultra-widefield OCT platform. The present study aimed to evaluate vitreous inflammation not only in the macula, however also in a wider area using wide-angle OCT (W-OCT).

While OCT allows evaluations that were previously unavailable, the method is not perfect. As a result, assessments must be made by combining a variety of tests. For example, the rapid kit for influenza offers a reported sensitivity of 27% and specificity of 97% [11]. When the sensitivity is low, diagnoses are made based on other tests and medical conditions. Ophthalmoscopy is the basis of examinations of the eye. However, it could be difficult to perform in some cases. OCT is an advanced examination and is thought to provide stable results in some difficult case.

Artificial intelligence (AI) is therefore being utilized in the identification of diseases including macular degeneration [12,13]. Automated analysis of results from OCT may allow more accurate diagnosis in the future. No reports appear to have described the sensitivity and specificity of W-OCT for identifying uveitis. This study therefore investigated the sensitivity and specificity of "Ophthalmoscopic findings" and "W-OCT" findings.

## Materials and methods

### Data collection

This retrospective study included patients who visited the Ocular Inflammation and Tumor Outpatient Clinic at Osaka Metropolitan University. Patients included in this study were from January 2019 to January 2022. All patients had been diagnosed with uveitis and had undergone W-OCT. Descriptions of anterior chamber inflammation and vitreous inflammation were extracted from Ophthalmoscopic findings. Inflammation results were based on the presence or absence of inflammatory cells. The existence of inflammatory cells ("Cells") and vitreous opacity ("VO") was determined by W-OCT. Wide-angle fluorescein angiography was also performed, with the results defined as "FAG findings". "cells" were defined as hyperreflective dots that were larger and greater density than noise in background [14]. "Vitreous opacity" were defined as hyperreflective area in background" Fig 1". Inflammatory findings such as vasculitis and pigment epitheliitis were examined under fluorescein angiography. Fluorescein

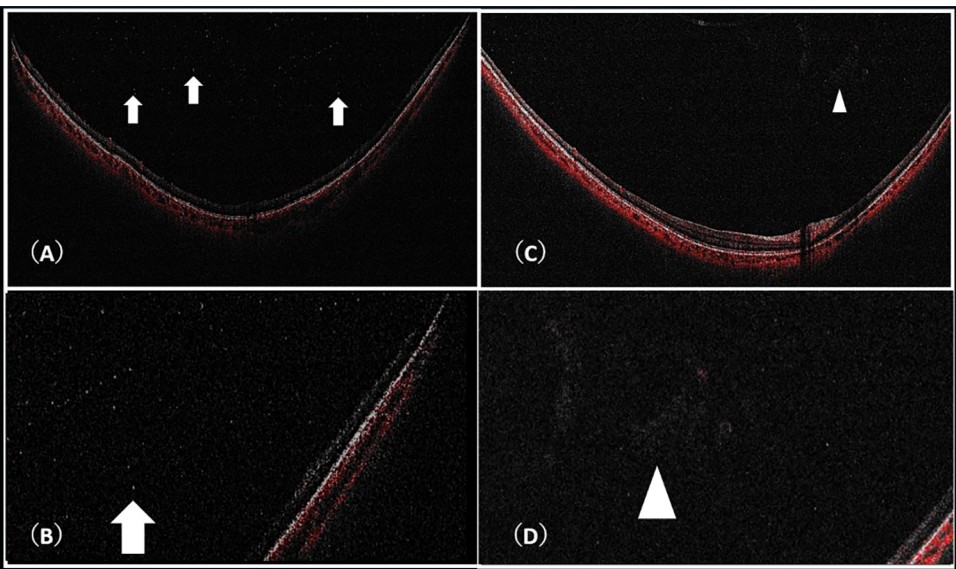

**Fig 1.** (A) Many cells are observed in the vitreous cavity. White arrow. (B) Enlarged image of (A). Magnify 2x. (C) Large vitreous opacity is seen in the vitreous cavity. White arrowhead. (D) Enlarged image of (C). Magnify 2x.

angiography was performed using a TRC-50DX (Topcon, Tokyo, Japan), Heidelberg Spectralis-OCT2 (Heidelberg Engineering, Heidelberg, Germany) or Optos 200Tx (Optos PLC, Dunfermline, Scotland) system. Each judgment was made as the consensus decision of three doctors (N.M, A.S, and M.T) in a specialized outpatient clinic. In cases of discrepancies, decision was made by consultation with S.H. The Xephilio OCT-S1 (Canon) is a swept-source W-OCT system that can capture 78˚ horizontal and 68˚ vertical. Tracking was used for the examination, and a wide-area imaging mode was used.

We obtained consent from all patients. This study complied with the principles of the Declaration of Helsinki and was approved by the Ethics Review Committee of Osaka Metropolitan University (approval no. **2019–062**). For minors, consent must be obtained from a parent or guardian.

## Statistical analysis

Sensitivity and specificity were calculated. Patient characteristics were summarized using descriptive statistics for quartiles and percentages, and the $\chi^2$ test was used for comparisons. McNemar's test was performed for the comparison of inflammatory findings. Statistical analyses were performed using SPSS Statistics version 22 software (IBM Japan, Tokyo, Japan). Values of $P < 0.05$ were considered statistically significant.

## Results

### Characteristics

In total, 132 eyes from 68 patients were included in the study "Table 1". The enrolled patients comprised 34 males and 34 females, with a mean age of 53.97±22.71years (range, 6–87 years). A detailed distribution of the 132 eyes is shown in "Table 1". Uveitis was the most frequently registered vitreous cells (54.5%), with no identified cause in any case. Sarcoidosis (12.1%) and Vogt-Koyanagi-Harada syndrome (VKH) (13.6%) were also common. Infectious uveitis was diagnosed by blood sampling and PCR (polymerase chain reaction) testing of anterior chamber fluid and vitreous fluid. Patient characteristics were analyzed by posterior inflammation examination

**Table 1. Characteristics of all patients.** IOP, ES, and age are shown as median and quartiles and were compared using the $\chi^2$ test.

| Factor | Group | n (%) |
|---|---|---|
| W-OCT Cells | None | 60 (45.5) |
| | Cells present | 72 (54.5) |
| W-OCT VO | None | 102 (77.3) |
| | VO present | 30 (22.7) |
| Ophthalmoscopic findings: Cells | None | 90 (68.2) |
| | Cells present | 42 (31.8) |
| Ophthalmoscopic findings: VO | None | 112 (84.8) |
| | VO present | 20 (15.2) |
| FAG findings | None | 30 (37.5) |
| | Inflammation present | 50 (62.5) |
| Eye | Right | 66 (50.0) |
| | Left | 66 (50.0) |
| Sex | Male | 45 (51.1) |
| | Female | 43 (48.9) |
| Disease | Acute retinal necrosis | 4 (3.0) |
| | Behcet disease | 2 (1.5) |
| | Cytomegalovirus | 2 (1.5) |
| | HTLV-1 | 2 (1.5) |
| | Sarcoidosis | 16 (12.1) |
| | Syphilis | 10 (7.6) |
| | Uveitis | 72 (54.5) |
| | VKH | 18 (13.6) |
| | Scleritis | 4 (3.0) |
| | Sympathetic uveitis | 2 (1.5) |
| | | Median [quartiles] |
| Intraocular pressure (IOP) | | 13.00 [11.00, 15.00] |
| Equivalent sphere (ES) | | -0.75 [-2.25, 0.25] |
| Age, years | | 57.50 [37.00, 72.25] |

VO: vitreous opacity, VKH: Vogt-Koyanagi-Harada syndrome.

"Table 2". IOP (intra ocular pressure) ($p = 0.026$) and age were significantly lower in patients in whom cells were present ($p = 0.006$). In "FAG findings", significant differences in IOP ($p = 0.022$) and age ($p = 0.013$) were seen between patients with and without inflammation.

## Relation to severity of uveitis

We examined the good and poor vision groups. The good visual acuity group was defined as LogMAR less than or equal to 0. The poor group was defined as greater than 0. There was no significant difference between the presence or absence of cells on W-OCT in both the good visual acuity and poor visual acuity groups(p = 0.888,0.511).

Similarly, there was no significant difference in VO on W-OCT between the good and poor visual acuity groups(p = 0.683,0.633).

## Sensitivity and specificity

Compared with "Ophthalmoscopic findings", for all cases "Table 3A", "W-OCT Cells" offered 51.1% sensitivity and 66.7% specificity, with detection of vitreous cells in posterior uveitis and

**Table 2. Characteristics by examination.** IOP, ES, and age are shown as median and quartiles. Statistical comparisons were performed using the $\chi^2$ test. Significant differences in IOP and ES were observed between W-OCT and FAG findings.

| | | Ophthalmoscopic findings: Cells | | | W-OCT Cells | | | FAG findings | | |
|---|---|---|---|---|---|---|---|---|---|---|
| None or Present | | None | Cells present | p-value | None | Cells present | p-value | None | Inflammation present | p-value |
| n | | 90 | 42 | | 60 | 72 | | 30 | 50 | |
| Eye (%) | Right | 43 (47.8) | 23 (54.8) | 0.575 | 30 (50.0) | 36 (50.0) | 1 | 14 (46.7) | 26 (52.0) | 0.818 |
| | Left | 47 (52.2) | 19 (45.2) | | 30 (50.0) | 36 (50.0) | | 16 (53.3) | 24 (48.0) | |
| Sex, n (%) | Male | 31 (51.7) | 14 (50.0) | 1 | 16 (40.0) | 29 (60.4) | 0.086 | 14 (66.7) | 17 (50.0) | 0.272 |
| | Female | 29 (48.3) | 14 (50.0) | | 24 (60.0) | 19 (39.6) | | 7 (33.3) | 17 (50.0) | |
| Disease, n (%) | Acute retinal necrosis | 3 (3.3) | 1 (2.4) | | 1 (1.7) | 3 (4.2) | | 2 (6.7) | 2 (4.0) | |
| | Behcet disease | 0 (0.0) | 2 (4.8) | | 0 (0.0) | 2 (2.8) | | 0 (0.0) | 2 (4.0) | |
| | Cytomegalovirus | 1 (1.1) | 1 (2.4) | | 1 (1.7) | 1 (1.4) | | 0 (0.0) | 0 (0.0) | |
| | HTLV-1 | 2 (2.2) | 0 (0.0) | | 0 (0.0) | 2 (2.8) | | 0 (0.0) | 0 (0.0) | |
| | Sarcoidosis | 10 (11.1) | 6 (14.3) | | 14 (23.3) | 2 (2.8) | | 2 (6.7) | 6 (12.0) | |
| | Syphilis | 5 (5.6) | 5 (11.9) | | 2 (3.3) | 8 (11.1) | | 5 (16.7) | 3 (6.0) | |
| | Uveitis | 46 (51.1) | 26 (61.9) | | 35 (58.3) | 37 (51.4) | | 13 (43.3) | 27 (54.0) | |
| | VKH | 17 (18.9) | 1 (2.4) | | 4 (6.7) | 14 (19.4) | | 5 (16.7) | 9 (18.0) | |
| | Scleritis | 4 (4.4) | 0 (0.0) | | 3 (5.0) | 1 (1.4) | | 2 (6.7) | 0 (0.0) | |
| | Sympathetic uveitis | 2 (2.2) | 0 (0.0) | | 0 (0.0) | 2 (2.8) | | 1 (3.3) | 1 (2.0) | |
| Intraocular pressure (IOP) | | 13.50 [11.00, 15.00] | 13.00 [11.00, 14.75] | 0.536 | 14.00 [12.00, 16.00] | 13.00 [11.00, 14.00] | 0.026* | 13.50 [12.00, 15.00] | 12.00 [10.00, 14.00] | 0.022* |
| Equivalent sphere (ES) | | -0.50 [-2.19, 0.47] | -1.19 [-2.25, 0.12] | 0.235 | -0.50 [-2.31, 0.28] | -0.94 [-2.25, 0.25] | 0.609 | -0.32 [-2.81, 0.84] | -1.44 [-4.06, -0.25] | 0.055 |
| Age, years | | 58.00 [45.75, 70.25] | 55.00 [31.75, 73.25] | 0.764 | 65.50 [52.75, 78.25] | 54.00 [30.25, 65.00] | 0.006* | 65.00 [56.00, 78.00] | 52.50 [32.75, 65.75] | 0.013* |

| | | Ophthalmoscopic findings: VO | | | W-OCT VO | | |
|---|---|---|---|---|---|---|---|
| None or present | | None | VO present | p-value | None | VO present | p-value |
| n | | 112 | 20 | | 102 | 30 | |
| Eye (%) | Right | 54 (48.2) | 12 (60.0) | 0.467 | 49 (48.0) | 17 (56.7) | 0.534 |
| | Left | 58 (51.8) | 8 (40.0) | | 53 (52.0) | 13 (43.3) | |
| Sex, n (%) | Male | 54 (48.2) | 11 (55.0) | 0.633 | 47 (46.1) | 18 (60.0) | 0.215 |
| | Female | 58 (51.8) | 9 (45.0) | | 55 (53.9) | 12 (40.0) | |
| Disease, n (%) | Acute retinal necrosis | 4 (3.6) | 0 (0.0) | | 1 (1.0) | 3 (10.0) | |
| | Behcet disease | 0 (0.0) | 2 (10.0) | | 0 (0.0) | 2 (6.7) | |
| | Cytomegalovirus | 1 (0.9) | 1 (5.0) | | 1 (1.0) | 1 (3.3) | |
| | HTLV-1 | 0 (0.0) | 2 (10.0) | | 1 (1.0) | 1 (3.3) | |
| | Sarcoidosis | 12 (10.7) | 4 (20.0) | | 14 (13.7) | 2 (6.7) | |
| | Syphilis | 8 (7.1) | 2 (10.0) | | 7 (6.9) | 3 (10.0) | |
| | Uveitis | 64 (57.1) | 8 (40.0) | | 61 (59.8) | 11 (36.7) | |
| | VKH | 18 (16.1) | 0 (0.0) | | 12 (11.8) | 6 (20.0) | |
| | Scleritis | 4 (3.6) | 0 (0.0) | | 3 (2.9) | 1 (3.3) | |
| | Sympathetic uveitis | 1 (0.9) | 1 (5.0) | | 2 (2.0) | 0 (0.0) | |
| Intraocular pressure (IOP) | | 13.00 [8.00, 35.00] | 13.00 [8.00, 31.00] | 0.453 | 14.00 [8.00, 35.00] | 12.50 [8.00, 20.00] | 0.116 |
| Equivalent sphere (ES) | | -0.75 [-10.50, 7.50] | -1.38 [-10.25, 7.62] | 0.241 | -0.75 [-10.50, 7.50] | -1.25 [-10.25, 7.62] | 0.849 |
| Age, years | | 56.00 [6.00, 87.00] | 58.00 [6.00, 87.00] | 0.553 | 56.00 [6.00, 87.00] | 57.00 [17.00, 80.00] | 0.961 |

**Table 3. a) Sensitivity and specificity compared to ophthalmoscopy in all cases.** b) Sensitivity and specificity for panuveitis cases. c) Sensitivity and specificity of "W-OCT Cells" compared to fluorescence for all cases and panuveitis.

| All cases | | | | | | |
|---|---|---|---|---|---|---|
| Examination | Sensitivity (%) | Specificity (%) | AUC | Positive likelihood ratio | Negative likelihood ratio | p-value |
| W-OCT Cells | 51.1 | 66.7 | 0.531 | 1.533 | 0.733 | 0.00014 |
| W-OCT VO | 78.6 | 30 | 0.578 | 1.122 | 0.714 | 0.144 |
| FAG findings | 52.9 | 89.7 | 0.71 | 5.118 | 0.525 | 0.000119 |
| b) | | | | | | |
| Panuveitis | | | | | | |
| Examination | Sensitivity (%) | Specificity (%) | AUC | Positive likelihood ratio | Negative likelihood ratio | p-value |
| W-OCT Cells | 44.7 | 68.8 | 0.558 | 1.432 | 0.804 | 0.00326 |
| W-OCT VO | 68.2 | 30 | 0.494 | 0.974 | 1.061 | 0.19 |
| FAG findings | 53.8 | 91.7 | 0.706 | 6.462 | 0.503 | 0.00555 |
| c) | | | | | | |
| W-OCT Cells v.s. FAG findings | | | | | | |
| Examination | Sensitivity (%) | Specificity (%) | AUC | Positive likelihood ratio | Negative likelihood ratio | p-value |
| All cases | 66.7 | 70 | 0.675 | 2.222 | 0.476 | 0.424 |
| Panuveitis | 53.3 | 78.3 | 0.668 | 2.453 | 0.598 | 0.773 |

panuveitis differing significantly ($p$ = 0.00014). In panuveitis "Table 3B", "W-OCT Cells" offered 44.7% sensitivity and 68.8% specificity, again showing a significant difference compared to ophthalmoscopy ($p$ = 0.00326).

Compared to "Ophthalmoscopic findings", for all cases "Table 3A", "W-OCT VO" offered 78.6% sensitivity and 30% specificity, showing no significant difference ($p$ = 0.144) in the detection of vitreous opacity. For panuveitis "Table 3B", sensitivity was 68.2% and specificity was 30%, showing no significant difference ($p$ = 0.19).

Comparing "W-OCT Cells" and "FAG findings", for all cases "Table 3C", sensitivity was 66.7% and specificity was 70%, showing no significant difference ($p$ = 0.424). In panuveitis "Table 3C", sensitivity was 53.3% and specificity was 78.3%, showing no significant difference ($p$ = 0.773).

## Discussion

"W-OCT Cells" showed low sensitivity and high specificity compared to "Ophthalmoscopic findings". Observing inflammatory cells in the near retina is difficult under ophthalmoscopy in normal practice. Even the smallest inflammatory cells are recorded by W-OCT, which is considered highly specific. However, W-OCT is considered more sensitive because it detects inflammation that is not apparent under ophthalmoscopy. OCT is more accurate than ophthalmoscopy for measurement because an objective assessment can be provided [15]. The present study showed that W-OCT may be more sensitive to inflammation than ophthalmoscopy. W-OCT can scan a wide area of retina, however does not capture the whole vitreous cavity. In anterior vitreous inflammation cases, OCT are easily identified by ophthalmoscopy in all cases.

This difference may have contributed to the significant differences in posterior inflammation. In other hands, pan uveitis may be easier to evaluate with W-OCT because of the wide range of inflammation. Therefore, if only cases of pan uveitis are examined, inflammatory cells may be detected better than ophthalmoscopic findings.

In addition, W-OCT Cells and FAG findings showed no significant difference in identifying in all cases and pan uveitis in Table 3C). This result suggested that FAG could be assessed retinal inflammation, which might not reflect posterior vitreous cavity inflammation.

In pan uveitis, inflammation is seen throughout the entire area, which may have reduced discrepancies by location. In some diseases, such as pan uveitis, W-OCT may be able to assess intraocular inflammation.

In terms of patient background characteristics, both "W-OCT Cells" and "FAG findings" showed significant differences according to IOP and age. Both groups with inflammation were younger with significantly lower IOP. In this study, "W-OCT cells" showed low sensitivity and high specificity, suggesting that some inflammatory findings may not be detected by W-OCT. In all cases, W-OCT and FAG findings were significantly different compared to ophthalmoscopic findings. This could represent bias due to the cases tested in this study.

A previous report described sarcoidosis as the most common cause of uveitis in Japan [16]. In our study, however, VKH was slightly more common than sarcoidosis. This may be due to the small overall number of participants and because W-OCT was not performed for all patients, resulting in bias. In Behcet's disease, both Cells and VO were observed in all test groups. This may be a result of the high prevalence of Behcet's disease, which could have been detected in all the tests. On the other hand, HTLV-1 cells were not observed as "Ophthalmoscopic findings" however were observed by "W-OCT". This suggests that W-OCT may be able to detect slight inflammation that cannot be detected by ophthalmoscopy. W-OCT allows direct recording of inflammatory cells, including on the periphery. Therefore, new disease names and classifications may be added in the future.

IOP was lower when W-OCT detected cells, however this was not observed when "Ophthalmoscopic findings" detected cells. Uveitis is known to increase IOP [17], however the underlying mechanisms are variable. On the other hand, inflammation in the eye decreases aqueous humor production, reducing IOP [18]. "W-OCT cells" group found significant IOP lowering, however not in "Ophthalmoscopic findings" group. W-OCT may capture inflammation more acutely than ophthalmoscopic findings in Table 2. In particular, W-OCT may capture inflammation in a wider area around the retina than can be observed with OCT. Therefore, it is possible that there is a significant difference due to the selection of a more inflammatory group. Almost all of the present cases were examined at the time of initial examination, and changes in corner angles thus may not yet have occurred due to the short duration of the disease.

OCT has become an important part of medical practice in dealing with uveitis [19,20]. In recent years, various methods have been reported to quantify uveitis [21,22]. However, past reports have been evaluated using OCT of the macula only. Laser flare photometry is useful for assessing inflammation in the anterior chamber [23]. Schalnus reported that anterior-chamber inflammation in laser flare relates to anterior uveitis [24], however not to posterior uveitis. The present study found no notable differences in sensitivity or specificity between all cases and cases with panuveitis. More recently, AI analyses have been conducted using OCT [25]. Since findings of inflammation can be stored in OCT data, further advances may be made by AI using deep learning for W-OCT images. Combining ophthalmoscopic findings with W-OCT may allow evaluation of inflammation in uveitis from anterior uveitis to posterior uveitis.

Fluorescein angiography also plays an important role in the diagnosis of uveitis [26]. Wide-angle examination is also important in fluorescein angiography [27]. In this study, no significant difference in findings was seen between "W-OCT" and "FAG findings". In anterior uveitis, fluorescence examinations are also useful [28], and anterior uveitis may also show inflammation in the vicinity of the retina. In the future, W-OCT could be used to evaluate peripheral retinal inflammation instead of fluorescence.

In terms of VO, sensitivity was high and specificity was low for both all cases and panuveitis cases. VO can be localized, and many cases were not identified by W-OCT. VO was also present in a small number of cases. For these reasons, VO showed no significant difference between ophthalmoscopy and W-OCT. Future improvements in imaging accuracy may make

VO evaluation more accurate and quantifiable. If VO can be accurately quantified, the intraocular inflammatory state and components of VO may be discernible.

"W-OCT" and "FAG findings" showed that patients with inflammation tended to have lower IOP and were younger. In uveitis, IOP is increased [29]. In normal eyes, IOP increases with age in populations from Europe [30] and the United States, however decreases in Japanese populations [31]. IOP is affected by a variety of factors, including refractive index and blood pressure [32]. Younger patients may be more sensitive to inflammation, and factors other than age may also contribute to lower IOP.

## Limitations

All participants in this study were Japanese, so effect of ethnicity cannot be ruled out. The study cohort was small, because few patients had undergone W-OCT. Due to the high scanning speed of W-OCT, different objects may potentially be captured and misidentified as cells. It has been reported that OCT in the anterior segment optical coherence tomography can reclassify artifacts as cells [33]. Therefore, it is possible that the artifacts may be used to identify cells in this study as well in W-OCT. This may be due to OCT image blurring by cataracts [34]. W-OCT can evaluate retinal conditions from the posterior vitreous. However, it cannot evaluate the anterior vitreous state. Since OCT images are affected by anterior chamber conditions, W-OCT evaluation may differ due to inflammation or cataracts in the anterior vitreous [35].

## Conclusions

W-OCT may be superior to ophthalmoscopy in detecting vitreous cells in uveitis. W-OCT alone has the potential to determine the severity and diagnosis of uveitis. With the development of future analytical programs, sensitivity and specificity may increase over ophthalmoscopic findings. W-OCT also showed similar acquisition of inflammatory findings compared to FAG. Therefore, intraocular inflammation may be assessable by W-OCT instead of FAG. W-OCT is an objective, reproducible method as a machine evaluation. W-OCT may create a new standard for assessing the status of intraocular inflammation.

## Supporting information

**S1 File.**
(XLSX)

## Acknowledgments

We are grateful to Sentaro Kusuhara, M.D., Ph.D., for his writing advice.

## Author Contributions

**Conceptualization:** Mizuki Tagami.

**Data curation:** Norihiko Misawa, Yusuke Haruna.

**Formal analysis:** Norihiko Misawa.

**Funding acquisition:** Shigeru Honda.

**Investigation:** Norihiko Misawa, Mizuki Tagami, Atsushi Sakai, Yusuke Haruna.

**Methodology:** Atsushi Sakai.

**Software:** Norihiko Misawa.

**Supervision:** Shigeru Honda.

**Validation:** Mizuki Tagami.

**Visualization:** Mizuki Tagami.

**Writing – original draft:** Norihiko Misawa.

**Writing – review & editing:** Mizuki Tagami.

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
