## [Decision Letter · Decision Letter 0]

8 Jan 2023

PONE-D-22-28716Relationship between ultra-widefield optical coherence tomography and ophthalmoscopy for detecting posterior inflammation in posterior uveitis and panuveitisPLOS ONE

Dear Dr. Tagami,

Thank you for submitting your manuscript to PLOS ONE. After careful consideration, we feel that it has merit but does not fully meet PLOS ONE’s publication criteria as it currently stands. Therefore, we invite you to submit a revised version of the manuscript that addresses the points raised during the review process.

We look forward to receiving your revised manuscript.

Kind regards,

De-Kuang Hwang

Academic Editor

PLOS ONE

Journal Requirements:

“No”

“no competing interests”

Additional Editor Comments:

1. please provide explain the definition of "cells" and "vitreous opacity" in W-OCT in Method.

2. please provide figure(s) to demonstrate the vitreous cells and vitreous opacity in W-OCT.

3. is there any correlation between "amount of vitreous cells" and "degree of vitreous opacity" in W-OCT, and clinical severity of uveitis?

Reviewers' comments:

Reviewer's Responses to Questions

**Comments to the Author**

1. Is the manuscript technically sound, and do the data support the conclusions?

Reviewer #1: Yes

Reviewer #2: Yes

2. Has the statistical analysis been performed appropriately and rigorously? 

Reviewer #1: Yes

Reviewer #2: Yes

3. Have the authors made all data underlying the findings in their manuscript fully available?

Reviewer #1: Yes

Reviewer #2: Yes

4. Is the manuscript presented in an intelligible fashion and written in standard English?

Reviewer #1: Yes

Reviewer #2: Yes

5. Review Comments to the Author

Reviewer #1: 1. line 73: 2 comma

2. line 197: Uveitis is known to increase IOP due to clogging of the trabecular meshwork with inflammatory cells, trabeculitis, pupillary block, synechial closure or secondary to steroid usage. However, decrease aqueous humor production may cause lower IOP initially. Lower IOP was observed in “W-OCT cells” group, yet was not observed in "Ophthalmoscopic findings " detected cells group. Try to clarify the reasons or mechanism in discussion.

3. line 206: They reported that anterior-chamber inflammation… Please list the author’s name.

Reviewer #2: Line 73: “Ophthalmoscopy is the basis of examinations of the eye, , could be difficult to be performed in some cases”. Please change to “Ophthalmoscopy is the basis of examinations of the eye. However, it could be difficult to perform in some cases.”

Line 86: “Patients were included in this study from January 2019 to January 2022.”, Please change to “Patients included in this study were from January 2019 to January 2022.”

Line 99: “decision was made by consultation S.H.” Please change to “decision was made by consultation with S.H.”

Line 124: Please change “Cells” to “cells”.

Line 135: Please correct the spelling of “Fluorescenc”

Line 173: The authors stated that “For identifying panuveitis, no significant difference existed between ophthalmoscopy and W-OCT.” Yet in line 145, the author stated in panuveitis (Table 3b), “W-OCT Cells” offered 44.7% sensitivity and 68.8% specificity, again showing a significant difference compared to ophthalmoscopy. Please explain the seemingly contradictory statements.

Please state and explain the likelihood of OCT misclassifying artifacts as cells either in this study or citing from other studies.

Since W-OCT cannot assess cells in the anterior chamber, W-OCT may be more of a tool used for posterior uveitis, instead of panuveitis, as a strong anterior chamber inflammation may introduce significant artifacts into posterior cells evaluation. You may need to include this statement in your limitations.

6. PLOS authors have the option to publish the peer review history of their article (what does this mean?). If published, this will include your full peer review and any attached files.

Reviewer #1: No

Reviewer #2: No

---

## [Author Response · Author response to Decision Letter 0]

27 Jan 2023

Additional Editor Comments:

1. please provide explain the definition of "cells" and "vitreous opacity" in W-OCT in Method.

P6 L91

“cells” were defined as hyperreflective dots that were larger and greater density than noise in background [14] . "vitreous opacity” were defined as hyperreflective area in background” Figure1”.

We added the above to the method article. Thank you for pointing out this critical part. Appreciate your advice greatly.

2. please provide figure(s) to demonstrate the vitreous cells and vitreous opacity in W-OCT.

P8L117

Figure1 

(A) Many cells are observed in the vitreous cavity. White arrow.

(B) Enlarged image of (A). Magnify 2x.

(C) Large vitreous opacity is seen in the vitreous cavity. White arrowhead.

(D) Enlarged image of (C). Magnify 2x.

We added figure to the method article. Thank you for your advice.

3. is there any correlation between "amount of vitreous cells" and "degree of vitreous opacity" in W-OCT, and clinical severity of uveitis?

P15 L162

Relation to severity of uveitis

We examined the good and poor vision groups. The good visual acuity group was defined as LogMAR less than or equal to 0. The poor group was defined as greater than 0. There was no significant difference between the presence or absence of cells on W-OCT in both the good visual acuity and poor visual acuity groups(p=0.888,0.511).

Similarly, there was no significant difference in VO on W-OCT between the good and poor visual acuity groups(p=0.683,0.633).

W-OCT can scan deep and wide areas quickly. However, it is difficult to compare the same location between patients, especially for the vitreous cavity. Also, because the scan is fast, more cells can be detected.So, accurate measurement of the cell number was difficult. Therefore, cell count and degree of vitreous opacity was not performed. The good vision group and the poor vision group were examined.No association was found with the presence or absence of cells or vitreous opacity.

There was no significant difference from clinical severity of uveitis.

We added figure to theresult article. Thank you for your advice.

Reviewer #1: 

1. line 73: 2 comma

P5L72

Ophthalmoscopy is the basis of examinations of the eye.

Mistakes have been corrected. Thanks for pointing out the details.

2. line 197: Uveitis is known to increase IOP due to clogging of the trabecular meshwork with inflammatory cells, trabeculitis, pupillary block, synechial closure or secondary to steroid usage. However, decrease aqueous humor production may cause lower IOP initially. Lower IOP was observed in “W-OCT cells” group, yet was not observed in "Ophthalmoscopic findings " detected cells group. Try to clarify the reasons or mechanism in discussion.

P19L235

"W-OCT cells" group found significant IOP lowering, however not in "Ophthalmoscopic findings" group. W-OCT may capture inflammation more acutely than ophthalmoscopic findings in table2. In particular, W-OCT may capture inflammation in a wider area around the retina than can be observed with OCT. Therefore, it is possible that there is a significant difference due to the selection of a more inflammatory group.

We added the above to the discussion article. Thank you for your advice.

3. line 206: They reported that anterior-chamber inflammation… Please list the author’s name.

P20L246

Schalnus reported that anterior-chamber inflammation in laser flare relates to anterior uveitis[24], however not to posterior uveitis.

Changed citation location and subject text. Thank you for pointing out the difficult to understand part.

Reviewer #2:

Line 73: “Ophthalmoscopy is the basis of examinations of the eye, , could be difficult to be performed in some cases”. Please change to “Ophthalmoscopy is the basis of examinations of the eye. However, it could be difficult to perform in some cases.”

Line 86: “Patients were included in this study from January 2019 to January 2022.”, Please change to “Patients included in this study were from January 2019 to January 2022.”

Line 99: “decision was made by consultation S.H.” Please change to “decision was made by consultation with S.H.”

Line 124: Please change “Cells” to “cells”.

Line 135: Please correct the spelling of “Fluorescenc”

P5L72

Ophthalmoscopy is the basis of examinations of the eye. However, it could be difficult to perform in some cases.

P6L84

Patients included in this study were from January 2019 to January 2022.

P6L98

In cases of discrepancies, decision was made by consultation with S.H.

P10L139

patients in whom cells were present (p=0.006).

P13L150

Significant differences in IOP and ES were observed between W-OCT and　FAG findings.

Thanks for pointing that out. I have corrected the error.

Line 173: The authors stated that “For identifying panuveitis, no significant difference existed between ophthalmoscopy and W-OCT.” Yet in line 145, the author stated in panuveitis (Table 3b), “W-OCT Cells” offered 44.7% sensitivity and 68.8% specificity, again showing a significant difference compared to ophthalmoscopy. Please explain the seemingly contradictory statements.

P18L201

In anterior vitreous inflammation cases, OCT are easily identified by ophthalmoscopy in all cases.

 This difference may have contributed to the significant differences in posterior inflammation. In other hands, pan uveitis may be easier to evaluate with W-OCT because of the wide range of inflammation. Therefore, if only cases of pan uveitis are examined, inflammatory cells may be detected better than ophthalmoscopic findings. 

In addition, W-OCT Cells and FAG findings showed no significant difference in identifying in all cases and pan uveitis in table 3C). This result suggested that FAG could be assessed retinal inflammation, which might not reflect posterior vitreous cavity inflammation. 

We added the above to the discussion article. Thank you for your advice.

Please state and explain the likelihood of OCT misclassifying artifacts as cells either in this study or citing from other studies.

P22L279

It has been reported that OCT in the anterior segment optical coherence tomography can reclassify artifacts as cells [33]. Therefore, it is possible that the artifacts may be used to identify cells in this study as well in W-OCT.

We added the above to the limitation article. Thank you for your advice.

Since W-OCT cannot assess cells in the anterior chamber, W-OCT may be more of a tool used for posterior uveitis, instead of panuveitis, as a strong anterior chamber inflammation may introduce significant artifacts into posterior cells evaluation. You may need to include this statement in your limitations.

P22L282

W-OCT can evaluate retinal conditions from the posterior vitreous. However, it cannot evaluate the anterior vitreous state. Since OCT images are affected by anterior chamber conditions, W-OCT evaluation may differ due to inflammation or cataracts in the anterior vitreous [35].

We added the above to the limitation article. Thank you for your advice.

---

## [Editor Report · Decision Letter 1]

31 Jan 2023

Relationship between ultra-widefield optical coherence tomography and ophthalmoscopy for detecting posterior inflammation in posterior uveitis and panuveitis

PONE-D-22-28716R1

Dear Dr. Tagami,

We’re pleased to inform you that your manuscript has been judged scientifically suitable for publication and will be formally accepted for publication once it meets all outstanding technical requirements.

Kind regards,

De-Kuang Hwang

Academic Editor

PLOS ONE
---

## [Editor Report · Acceptance letter]

2 Feb 2023

PONE-D-22-28716R1 

Relationship between ultra-widefield optical coherence tomography and ophthalmoscopy for detecting posterior inflammation in posterior uveitis and panuveitis 

Dear Dr. Tagami:

I'm pleased to inform you that your manuscript has been deemed suitable for publication in PLOS ONE. Congratulations! Your manuscript is now with our production department. 

Kind regards, 

on behalf of

Dr. De-Kuang Hwang 

Academic Editor

PLOS ONE